# Antifungal Effects and Potential Mechanism of Essential Oils on *Collelotrichum gloeosporioides* In Vitro and In Vivo

**DOI:** 10.3390/molecules24183386

**Published:** 2019-09-18

**Authors:** Dan Wang, Jing Zhang, Xiaoman Jia, Li Xin, Hao Zhai

**Affiliations:** Shandong Institute of Pomology, Tai’an 271000, Shandong, China; geum307@126.com (D.W.);

**Keywords:** *Collelotrichum gloeosporioides*, sweet cherry, essential oil, chemical composition, antifungal mechanism

## Abstract

The development of natural essential oil as an alternative to synthetic chemicals in the control of postharvest decay is currently in the spotlight. In the present study, the efficacy of seven essential oils in suppressing *Collelotrichum gloeosporioides* identified from sweet cherry was evaluated in vitro and clove oil was proved to be the most promising inhibitor. Thus, the antifungal properties and potential mechanisms of clove oil in vitro and in vivo by fumigation and contact treatments were intensively investigated. For *C. gloeosporioides*, the minimal inhibitory concentrations (MIC) of clove oil in air and contact phase were 80 and 300 μL/L in vitro testing, respectively. Based on the radial growth of *C. gloeosporioides* mycelium in medium, the fumgitoxic ability of essential oil was observed in a dose-dependent manner, which was not as dramatic as that under in vivo conditions. Furthermore, scanning electron microscopy and transmission electron microscopy of *C. gloeosporioides* exposed to clove oil exhibited obviously deleterious morphological and ultrastructural alterations confirming the disruption of fungal cell wall and endomembrane system, which resulted in increasing in permeability and causing the loss of intracellular constituents. In future, essential oils, combined with nano-emulsification approaches, could be good candidates as safe and effective antifungal agents for fungal spoilage of fresh commodities.

## 1. Introduction

Sweet cherries (*Prunus avium L.*) are highly appreciated for their vivid color, unique flavor, and nutritional value. They are currently the earliest mature deciduous fruit tree cultivated in northern China, and become one of the fastest growing fruits in Chinese and international markets [1]. However, sweet cherries are juicy, have thin skin with a susceptibility to mechanical damage. Additionally, their harvesting season has high temperature and humidity, so that the growth rate of pathogenic fungi latent in the pre-harvest, picking and transport is accelerated, eventually leading to fruit spoilage and loss of commercial value [2,3]. Among them, *Collelotrichum gloeosporioides (C. gloeosporioides)* is one of the major and serious saprophytic pathogens of sweet cherries causing postharvest losses at a high frequency [4].

Although the control of postharvest diseases in various fruits and vegetables currently still relies mainly on the application of chemical fungicides, development of resistance to some fungicides and an increase in consumer concern regarding carcinogenic impacts, residual toxicity and environmental pollution has transformed attention of researchers to naturally derived antifungal alternatives to synthetic compounds [5,6]. Plant essential oil is a mixture of aromatic components such as terpenes, aldehydes, esters and alcohols extracted from different plant organs. The main modes of its application include fumigation, spraying and soaking [7,8]. Essential oils have not only been shown to possess a strong and wide antimicrobial and antifungal spectrum, but also act synergistically among many compounds and exhibit multi-antimicrobial targets which lead to low resistance-inducing effects [9]. Therefore, essential oils are one of the most promising groups of natural compounds for the development of efficient antimicrobial agents.

In recent years, the microbial inhibitory effect of essential oils and their major antifungal compounds against *C. gloeosporioides* has been partially researched, and it has been proved that thyme oil and *Mentha piperita L*. essential oil, as well as thymol and R-(-)-carvone exhibited strong antimicrobial activity to control anthracnose during the postharvest period [10,11,12,13]. Nevertheless, systemic and in-depth studies on postharvest rot pathogens of sweet cherry, especially on *Colletotrichum* species are lacking. Hence, the objective of this study was to assess the in vitro effectiveness of several essential oils in controlling *C. gloeosporioides* isolated from sweet cherry by fumigation and contact treatments; then, with the best essential oil obtained in vitro, to evaluate decay inhibition not only in artificially inoculated and wounded fruits but also in unwounded fruits (in vivo), and to reveal the antifungal mechanism clarified by the effects of essential oil on the hyphal morphology and cellular ultrastructure and changes of fungal cell wall and cell membrane of *C. gloeosporioides* in the presence of essential oil. This investigation is helpful for providing theoretical basis and technical support for commercial application of plant essential oils in the green control of fungal diseases of fresh fruits and vegetables.

## 2. Results and Discussion

### 2.1. The Morphological Characteristics of C. Gloeosporioides

Anthracnose, caused by *C. gloeosporioides*, is one of the most aggressive diseases during the postharvest of sweet cherries and limits good fruit quality [14]. As shown in Figure 1, a strain of fungus was isolated from sour rot sweet cherry, which was identified as *C. gloeosporioides* after preliminary evaluation [15].

### 2.2. Antimicrobial Effect of Essential Oils In Vitro

#### 2.2.1. Antifungal Activity of Seven Essential Oils In Vitro

The inhibitory effects of seven typical essential oils against *C. gloeosporioides* after two days’ exposure are presented in Table 1. In general, the antifungal activities at 100 μL/L concentration in gas diffusion were equal to that at 1000 μL/L concentration in solid diffusion except palmarosa and lemon oil, which was also proved that the high lipophilic nature of mycelia coupled with a large surface area relative to the volume of a fungus, vapours of essential oils may act mainly by accumulation on mycelia than in the agar [16]. Moreover, cinnamon and clove oil exhibited strong antifungal action both in vitro fumigation and contact assay (*p* < 0.05), which could inhibit the radial growth of *C. gloeosporioides* completely at 100 μL/L (vapour phase), 1000 μL/L (contact phase), respectively. Besides, thyme oil possessed relatively good inhibitory effects on *C. gloeosporioides (p* < 0.05). As previously reported, thyme oil was proved to be the most effective inhibitor in control of anthracnose diseases [17,18,19]. In contrast, the performance of clove oil was not as good as present testing, requiring at least 3000 and 2000 μL/L (contact phase) for total inhibition of two strains of *C. gloeosporioides*, respectively [20,21]. Comparison between our results and those of other reports exhibited difference, probably due to plant species or sites, plant organs extracted, as well as the harvesting time [22]. Therefore, it is of reference value to analyze its composition before subsequent experiments.

#### 2.2.2. Chemical Composition and Antifungal Activity of Optimized Essential Oils In Vitro

Based on the above results, we selected two best-performing cinnamon and clove oil. The chemical composition of them was analyzed by GC-MS and shown in Figure 2 and Table 2. Analysis results showed that there are three major kinds of compounds in clove oil we used, respectively, eugenol (10.261 min), β-caryophyllene (11.246 min) and eugenol acetate (12.336 min). The contents of benzaldehyde (4.578 min) and trans-cinnamaldehyde (9.214 min) were significantly higher than other components in cinnamon oil we used.

Subsequently, we further evaluated their fungistatic action by fumigation and contact treatments. The results shown in Figure 3 revealed that cinnamon oil has poorer effectiveness than clove oil at lower concentration by both application methods; however, there was no significant difference between them at higher concentration, which may be owing to their different qualitative and quantitative components with the greatest fungitoxic ability as well as their evaporation rate [23]. Some studies suggested that the concentration of exposure to essential oil was a dependent factor for antifungal activity under in vitro condition [24,25]. Our experiments also revealed that the correlations between inhibition rate and concentration were greater than 0.9, indicating that the inhibition ratio was proportional to concentration (Table 3, Figure 4). Also, as presented in Table 3, fumigation exhibited obviously stronger antifungal activity compared with contact assay from EC_50_ and MIC values of cinnamon and clove oil on *C. gloeosporioides*. The results further support the study, which found that the volatile assay was more effective than the contact assay in vitro, though both of them significantly reduced the fungi in comparison with the control [26,27].

### 2.3. The Control of Postharvest Fungal Decay of Sweet Cherries by Clove Oil In Vivo

#### 2.3.1. Efficacy of Clove Oil on Natural Decay Development in Intact Sweet Cherry Fruits

As shown in Figure 5, the incidence of infection in all essential oil-treated sweet cherries was significantly lower than that in the control at 26 °C for 6 days (*p* < 0.05). The percentage of decayed sweet cherries fumigated by clove oil at 100 μL/L was reduced by 44% compared to the control. Within the concentration of 100 μL/L, the higher the concentration of clove oil, the lower the disease incidence exhibited, whereas the treatment with 120 μL/L clove oil suppressed the decreasing trend. This may be related to the high concentration of essential oil, which would disorder the metabolic pathways associated with redox of the fruits, resulting in a reduction of resisting infection by pathogens [28]. Arras and Usai also showed that thyme oil was less effective on orange fruits than in vitro against *Penicillium digitatum* at higher concentration [29]. In practical application, the efficacy of essential oil on specific control of postharvest fruits decay is further dependent on the natural resistance and their desirable storage time of the fruit’s cultivars involved [30]. Furthermore, combining other bioactive natural compounds or technologies such as chitosan coating and pickering type immobilized film rendering essential oils more effective in creating conditions conducive to extending the storage life of the fresh produce, while maintaining the overall fruit quality, nutritional compounds and consumer acceptance [31,32].

#### 2.3.2. Efficacy of Clove Oil on Rot Development in Artificially Inoculated Sweet Cherry Fruits

Figure 5 showed an obvious antiseptic effect on inoculated sweet cherry fruits among treatments of essential oil and control, as well as a decreasing trend of rot rate within the concentration range of 20–80 μL/L. However, visible injury of essential oil on sweet cherries emerged at the concentration of 100 μL/L. Additionally, a statistically significant difference was also observed between unwound and wound at the same treated concentration (*p* < 0.05).

### 2.4. Observing the Effect of Clove Oil on Hyphal Morphology and Cellular Ultrastructure

SEM could intuitively reflect the morphological changes of *C. gloeosporioides*. In the control samples (Figure 6A,C), the hyphae was uniform and robust with a plump and smooth surface, while the treated mycelia (Figure 6B,D) became shrank, collapsed and winding following contact with clove oil at a concentration of 400 μL/L.

The effects of clove oil on the ultrastructure of *C. gloeosporioides* observed by TEM are shown in Figure 6E,F. The control revealed a typical fungal ultrastructure of which all organelles had normal and regular appearance, and were clearly identified. After treatment with clove oil, the structures of most organelles were indistinct and unidentifiable with deformed and disorganized mitochondria. The major changes included cell wall and endomembrane system. In treated fungi, the cell wall showed obvious thickening, becoming rough and villous, and the cell membrane was partially detached from the cell wall.

### 2.5. Effect of Clove Oil on the Cell Wall and Cell Membrane

Some researches demonstrated that the hydrophobicity of essential oil enabled them to partition in the lipids of the microorganism cell membrane and mitochondria, thereby increasing their permeability and leading to releasing intracellular constituents [33,34], and interfering with many biological processes [35]. In present study, it was found that the absorbance at 260 mm of the groups treated with 200, 300, 400 μL/L clove oil was higher than that of control and 100 μL/L treatment, suggesting that high concentration of essential oil led to the membrane permeability increased and nucleic acid released from *C. gloeosporioides* (Figure 7). However, no significant difference was observed among that of 200, 300, 400 μL/L clove oil, which was not completely consistent with the results of Shao et al. (2013) who observed a continuously rising trend of absorbance at 260 nm, possibly attributed to less yield of mycelia due to the growth inhibition as the concentration of essential oil increasing. The results from Figure 7 (bars) revealed that the protein content of fungal cells was markedly decreased as the amount of essential oil increased (100–400 μL/L), which also indicated the disruption to the permeability barrier of cell membrane structures [36]. 

## 3. Materials and Methods

### 3.1. Sweet Cherries Fruits and Essential Oils 

“Van” sweet cherries were used as test materials in the study which were hand-harvested from commercial sweet cherry plants in Tai’an, China, transported to the laboratory and sorted manually for uniform maturity and absence of blemishes. Fruits were stored at conditions of 0 ± 1 °C and 90–95% relative humidity.

Pure-grade essential oils of clove, thyme, palmarosa, lemongrass, lemon, cinnamon, and laurel leaf were purchased from Guangzhou Hengxin Spice Co., Ltd., Guangzhou, China, and stored in the dark, at room temperature.

### 3.2. Chemical Composition of Essential Oils Analyzed by GC-MS

The GC-MS analysis of cinnamon essential oil refers to the method of Zeng et al. [37].

### 3.3. Fungi and Culture

*C. gloeosporioides* was isolated from naturally decayed sweet cherry fruits and cultured on potato dextrose agar (PDA) medium at 26 °C for 7 days prior to experimentation.

### 3.4. Assessment of the Effect of Essential Oils on Mycelium Growth by Vapour and Contact Phase In Vitro

#### 3.4.1. Fumigation Assay

The antifungal activity was analyzed based on the modified method of Feng et al. [26]. For determination of vapor effect, glass petri plates (75 × 20mm in diameter, which offer 37.5 mL air space after addition of 12.5 mL PDA) were inoculated with 7 mm mycelia plug from 7-days-old culture. A sterilized filter paper disc (30 mm diameter) was attached on the center of the lid inner surface with different amount (1.5–5.5 μL/plate) of essential oils added onto the paper and then the plate was quickly covered. The controls were prepared similarly with the exception of the oil treatment. All plates were kept in an inverted position and incubated at 26 °C for 4 days. 

#### 3.4.2. Contact Assay

To determine contact effect, appropriate amounts of pure essential oil was dispersed using 10mL Tween 80 (0.2% *v*/*v*) and added to 50 mL PDA media (15 mL/plate finally) immediately before it was poured into the glass petri dishes (75 × 20 mm in diameter), to obtain final concentration of 100–500 μL/L. The controls received the same amount of Tween 80 mixed with PDA. *C. gloeosporioides* was inoculated by plating in the center of each plate with a mycelia streak of the fungus obtained from 7-days-old actively growing cultures. All plates were kept in an inverted position and incubated at 26 °C for 4 days.

The lowest concentration of the essential oil at which there was no strain growth for 48h was selected as the minimum inhibitory concentration (MIC). The percentage of fungal inhibition was calculated using the following equation:Inhibition rate (%) = (D_c_ − D_t_)/(D_c_ − D_0_) × 100(1)
where D_0_ is the diameter of initially inoculated plug, D_c_ is the diameter of control and D_t_ is the diameter of each treatment. The assay was performed three times.

### 3.5. Control of Fungal Decay of Sweet Cherries by Clove Oil In Vivo

#### 3.5.1. Effects of Clove Oil on Natural Infection Development in Intact Sweet Cherry Fruits

Experimental units consisted of 80 fruits arranged in plastic baskets (45 × 30 × 10 cm) and a 0.025 mm polyethylene (PE) bag that was left unsealed. Each treatment was replicated three times. Different amounts of clove oil were added to filter paper discs which were previously placed on the center of inner surface of PE bag, to obtain concentration of 20, 40, 80, 100, 120 μL/L. The control comprised filter paper discs without essential oil. All treated sweet cherries were stored at 26 °C. The rate of rot fruits was recorded after 6 days of incubation.

#### 3.5.2. Effects of Clove Oil on Rot Development in Artificially Inoculated and Wounded Fruit

Sweet cherries were wounded with a sterile puncher to make one uniform 2-mm deep by 6-mm wide wound on their peel at the equatorial region. A mycelium plug (5 mm diameter) was inoculated into the stabbed site of the fruit with inoculation of sterile PDA medium as control, which were wrapped by fresh-keeping film. The following operations were the same as Section 3.4.2.

### 3.6. Scanning Electron Microscopy (SEM) and Transmission Electron Microscopy (TEM) of Hyphal Morphology and Cellular Ultrastructure

Seven-day-old *C. gloeosporioides* were transferred into the centre of PDA medium and cultivated at 26 °C for 3 days. 1mL of conidial suspension (1 × 10^7^/mL) was added to shake flask which was contained 50 mL potato dextrose broth (PDB) medium. The suspension was incubated at 25 °C and 150 rpm for 3 days. The fully emulsified clove oil by Tween 80 solution (0.2% *v*/*v*) was incorporated to the shake flask to achieve the concentration of 450 μL/L (1.5MIC) against *C. gloeosporioides*, the group with no essential oil served as control. All of the samples were incubated at 25 °C and 150 rpm for 2 h and centrifuged at 4000 rpm for 10 min. Then mycelia collected were washed three times with 0.05 M (pH 7.0) phosphate buffer solution (PBS).

For SEM observation, the mycelia (2 mm^3^) was fixed with 2.5% glutaraldehyde and 4% formaldehyde for 4 h, washed with 0.1 M (pH 7.2) PBS, and dehydrated with gradient ethanol solution (30, 50, 70, 80, 90 and two times at 100% *v*/*v*). Subsequently, the samples were subjected to vacuum drying and coated with a layer of gold. All samples were visualized in a SEM (Sigma 300, ZEISS).

For TEM observation, the mycelia (1 mm^3^) was dipped into 3% glutaraldehyde, washed with 0.1 M (pH 7.2) PBS, and post fixed with 1% osmium tetraoxide, washed again with PBS. Afterward, the same dehydrated steps were carried out and the speciments were embedded in epoxy media. Then, the blocks were sectioned with a diamond knife into ultrathin sections (approximately 80 nm in thickness) using an ultratome (LKB-V, Dnn, Sweden). The ultrathin sections were contrasted with uranyl acetate followed by lead citrate for 30 min and examined in a TEM (MORADA-G2, Olympus, Japan).

### 3.7. Assessment of the Effect of Clove Oil on the Cell Wall and Cell Membrane

For emulsification, clove oil was added to Tween 80 solution (0.2% *v*/*v*) and make it fully emulsified. The emulsified essential oil was separately incorporated to a shake flask containing 50 mL PDB medium to achieve the concentrations of 100, 200, 300, 400 μL/L on *C. gloeosporioides*, the group without essential oil as control. 0.1 mL of conidial suspension of *C. gloeosporioides* (1 × 10^7^/mL) was added to each shake flask. The suspension was incubated at 28 °C and 200 rpm for 7 days. Samples collected were centrifuged at 4000 rpm for 15 min to obtain supernatant and the absorbance at 260 mm was measured in an UV/Vis spectrophotometer.

Meanwhile, hyphase precipitated above with the same weight was ground and centrifuged at 4000 rpm for 5 min to obtain supernatant. The protein content in supernatant was determined by coomassie blue staining.

### 3.8. Statistical Analyses

The data were expressed as the mean value ± standard deviation, and means were separated by the Tukey’s multiple range test when ANOVA was significant (*p* < 0.05).

## 4. Conclusions

The results of this study showed that clove oil was seen to be the most active inhibitor against *C. gloeosporioides* tested in vitro and in vivo by different treatments, and its fungitoxic activities may be attributed to the destruction of the cell wall and cell membrane, and the leakage of intracellular constituents. The lower EC_50_ and MIC values indicated that essential oil would be faster and more efficient in preventing fungal infection when used as gas than contact. However, direct application of essential oil on fruit surfaces by fumigation was easily subjected to fruit injury and quality deterioration due to their rapid or uncontrolled evaporation. Therefore, it is suggested that essential oil incorporated into emulsion system would be a promising method to prolong and improve their antimicrobial effect.

## Figures and Tables

**Figure 1 molecules-24-03386-f001:**
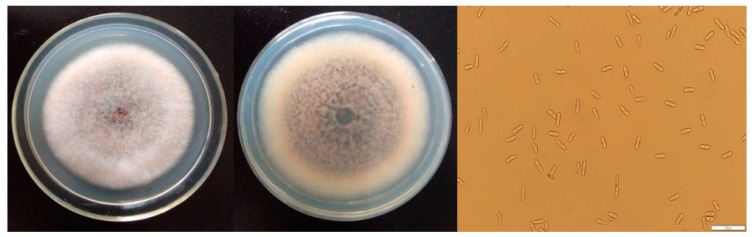
The colony morphology and conidia (after five days incubation) of *C. gloeosporioides* identified from sweet cherries.

**Figure 2 molecules-24-03386-f002:**
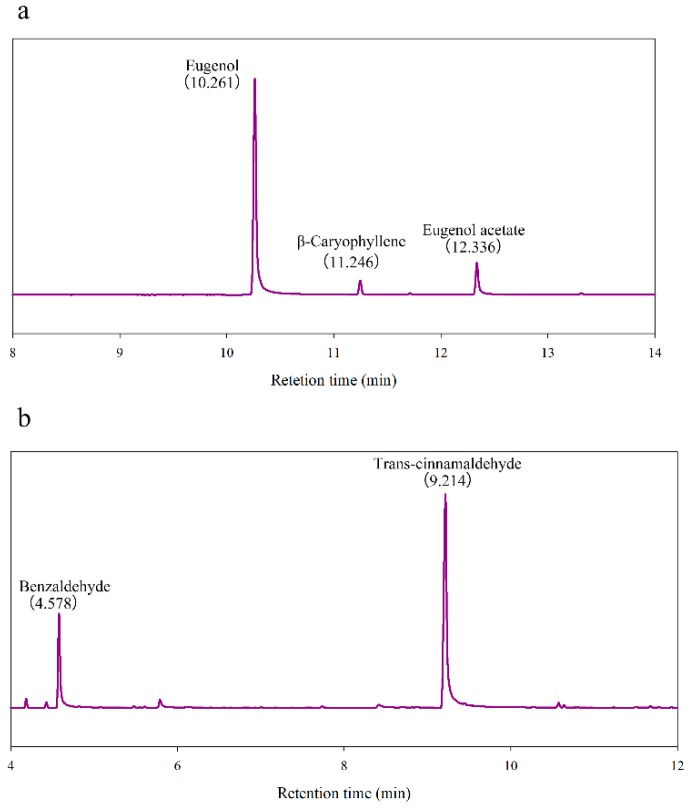
Chemical composition of essential oils analyzed by GC-MS: (**a**) clove oil; (**b**) cinnamon oil.

**Figure 3 molecules-24-03386-f003:**
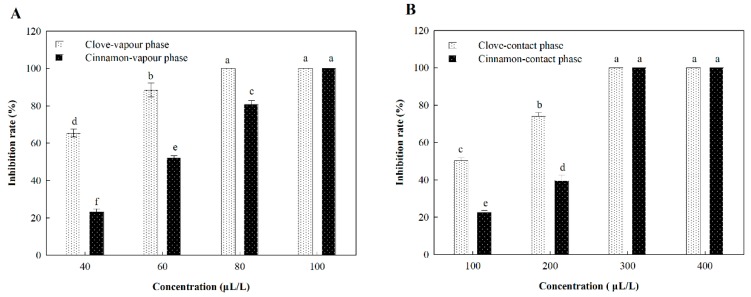
The inhibition activity of cinnamon and clove oil against *C. gloeosporioides* by (**A**) fumigation and (**B**) contact. Note: Different lowercase letters indicate a significant difference at the 5% level.

**Figure 4 molecules-24-03386-f004:**
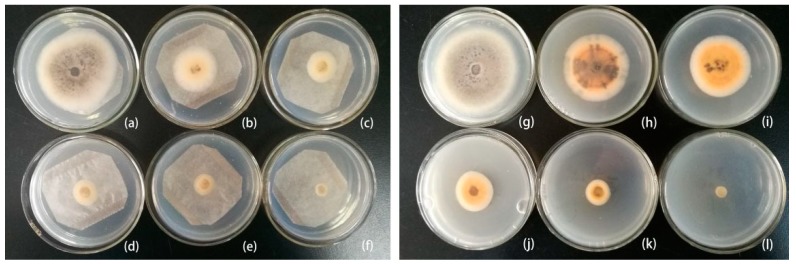
Radial growth of *C. gloeosporioides* mycelium treated with clove oil at 40 μL/L (**b**), 60 μL/L (**c**), 80 μL/L (**d**), 100 μL/L (**e**), 120 μL/L (**f**) compared with the control (**a**) in vapour phase and 100 μL/L (**h**), 200 μL/L (**i**), 300 μL/L (**j**), 400 μL/L (**k**), 500 μL/L (**l**) compared with the control (**g**) in contact phase after 4 days at 26 °C.

**Figure 5 molecules-24-03386-f005:**
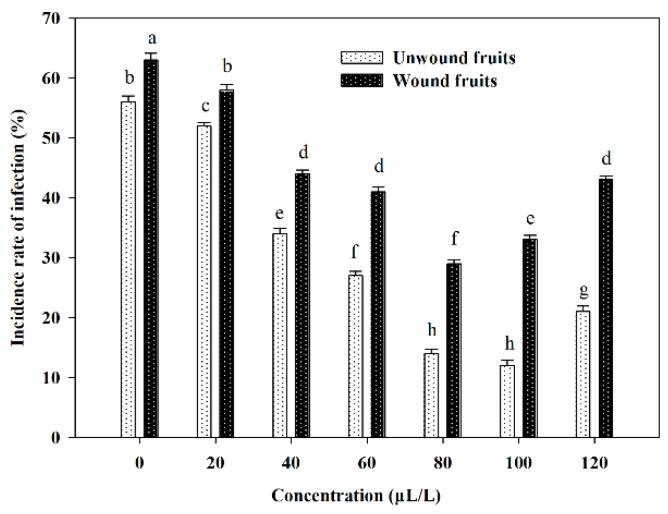
Inhibition effect on postharvest infection in wound and unwound sweet cherry fruits by clove oil. Note: Different lowercase letters indicate a significant difference at the 5% level.

**Figure 6 molecules-24-03386-f006:**
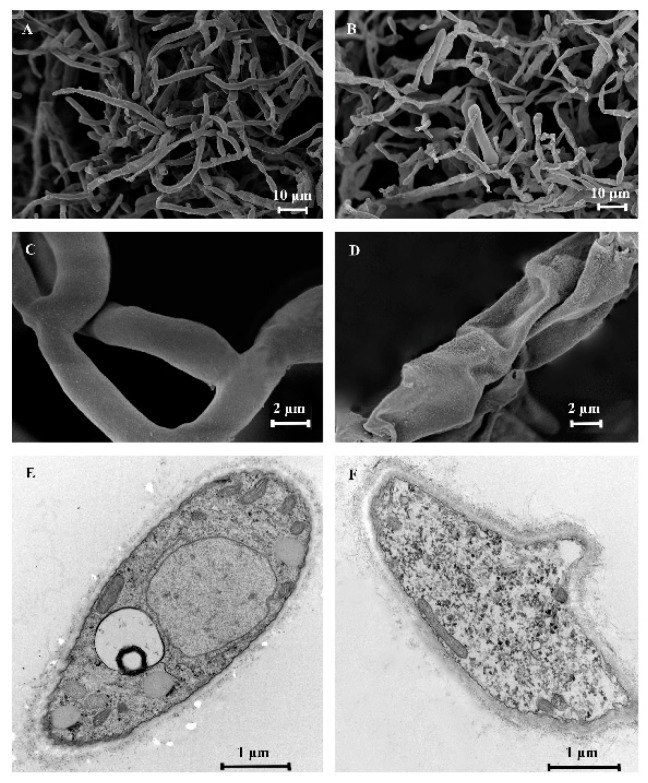
SEM of hyphal morphology (**A**–**D**) and TEM of hyphal ultrastructure (**E**,**F**) exposed to clove oil. (**A**,**C**,**E**) control, (**B**,**D**,**F**) treated with clove oil contact at 400 μL/L.

**Figure 7 molecules-24-03386-f007:**
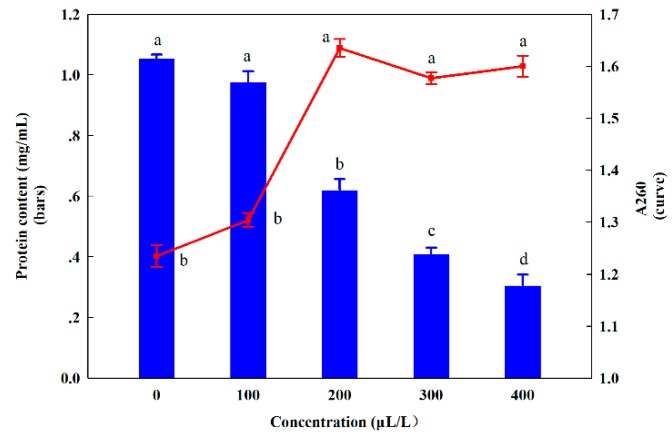
Effect of clove oil on nucleic acid leakage and protein content of *C. gloeosporioides*. Note: Different lowercase letters indicate a significant difference at the 5% level.

**Table 1 molecules-24-03386-t001:** The inhibition activity of essential oils against *C. gloeosporioides*.

Essential Oils	Rate of Mycelial Inhibition/%	
	Vapour Phase (100 μL/L)	Contact Phase (1000 μL/L)
Clove *Engenia caryophyllus*	100 ± 0.00 ^a^	100 ± 0.00 ^a^
Thyme *Thymus serpyllum*	73.24 ± 0.17 ^b^	85.43 ± 0.05 ^b^
Palmarosa *Cymbopogon martinii*	39.44 ± 0.13 ^c^	66.23 ± 0.32 ^c^
Lemongrass *Cymbopogon citratus*	44.37 ± 0.18 ^c^	40.40 ± 0.24 ^d^
Lemon *Citrus limonum*	3.52 ± 0.08 ^e^	11.35 ± 0.11 ^e^
Cinnamon *Cinnamomum cassia*	100 ± 0.00 ^a^	100 ± 0.00 ^a^
Laurel leaf *Laurus nobilis*	33.80 ± 0.10 ^d^	37.59 ± 0.08 ^d^

Note: Values are given as mean ± SD (n = 4). Different superscript letters in the same column indicates significant differences (*p* < 0.05).

**Table 2 molecules-24-03386-t002:** The main components of the essential oils of clove and cinnamon oil analyzed by GC–MS.

Peak No.	RT/min	Compounds	Content (%)
Clove oil			
1	10.261	Eugenol	78.952
2	11.246	β-caryophyllene	2.149
3	11.713	α- caryophyllene	7.261
4	12.336	Eugenol acetate	17.892
5	13.311	Oxetene	0.281
Cinnamon oil			
1	4.426	Terpene	0.957
2	4.578	Benzaldehyde	26.805
3	5.476	4-isopropyltoluene	0.486
4	5.550	Limonene	0.060
5	5.606	Amine leaf alcohol	0.102
6	5.791	Salicylaldehyde	3.260
7	6.639	Furfural	0.032
8	7.737	2-propanol	0.406
9	8.687	O-methoxybenzaldehyde	0.083
10	8.875	Phenylacetate	0.480
11	8.999	Trans-2-nonenal	0.036
12	9.214	Trans-cinnamaldehyde	67.247
13	9.830	2,4-nonadienal	0.038
14	10.414	2-undecenal	0.011

**Table 3 molecules-24-03386-t003:** EC_50_ and minimal inhibitory concentrations (MIC) values of cinnamon and clove oil on *C. gloeosporioides*.

Treatments	Virulence Regression Equation	EC_50_ (μL/L)	MIC (μL/L)
Clove oil-vapour phase	Y = 0.8677X + 32.508 R^2^ = 0.9643	20	80
Cinnamon oil-vapour phase	Y = 1.2975X − 26.775 R^2^ = 0.9913	59	100
Clove oil-contact phase	Y = 0.2482X + 25.147 R^2^ = 0.9992	100	300
Cinnamon oil-contact phase	Y = 0.3867X − 23.193 R^2^ = 0.9062	189	300

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
