# Peer review of "Antifungal Effects and Potential Mechanism of Essential Oils on Collelotrichum gloeosporioides In Vitro and In Vivo"

_molecules, 2019, doi:10.3390/molecules24183386_

Round 1

Reviewer 1 Report

The paper reports a classical procedures to evaluate the biological activities of EO on Collelotrichum gloeosporioides.

The paper reports radial diffusion, and GC profiles. The method is not validated and the quantitative results could not be reproducibile in absence of these information.

The EO analysis was focused merely on 2/3 compounds, while it could be represented by a larger number of compounds that could show biological activities.

Figures need to be imrproved in terms of quality and resolution.

Tables need to be checked for figure of merits

No vaucher speciments were reported

Author Response

Dear reviewer:

We greatly appreciate all the critiques and comments from you. Those comments are extremely helpful for us to improving our paper, and they provide valuable guidance for our future study. According to these comments, we have carefully improved our manuscript, and all the revisions are highlighted in red in the text. Please see below point-by-point responses to the reviewers’ comments:

 Response to Reviewer 1’s comments

Question 1:

The paper reports radial diffusion, and GC profiles. The method is not validated and the quantitative results could not be reproducible in absence of these information.

R: Thanks for your valuable comment. The radial diffusion (Fig.4) and GC profiles (Fig.2), only as intuitive display or qualitative results, combining with the corresponding quantitative outcomes Table 3 (the prior Table 2) and Table 2 (newly added), jointly together verify these results.

Question 2:

The EO analysis was focused merely on 2/3 compounds, while it could be represented by a larger number of compounds that could show biological activities.

R: Thanks for your valuable comment. As suggested, we have added Table 2 representing multiple compounds of the two essential oils. 

Question 3:

Figures need to be improved in terms of quality and resolution.

R: Thanks for your valuable comment. The figures enclosed in our previous submission are in TIFF format with the resolution of more than 300 dpi. Now there is no space to improve.

Question 4:

Tables need to be checked for figure of merits.

R: Thanks for your valuable comment. I think you mean Figure 4 and the prior Table 2, and it should be noted that Figure 4 showed radial growth of C. gloeosporioides mycelium treated with clove oil after 96h, while the values of MIC and EC50 of Table 3 (the prior Table 2) were defined as the concentration of no and half of strain growth for 48h.

Question 5:

No vaucher speciments were reported.

R: Thanks for your valuable comment. I am sorry that I can't understand what you mean.

Reviewer 2 Report

Abstract

Please, check the term “comiditities” 

Introduction:l

ine 27 Prunus, instead of prunus

Line 52 and line 54: in vitro should be written in italics

Line 64: limits instead of iimits

Results and discussion

Line 91 antifungal instead of Antifungal

Line 94: delete “are”

Line 102: please, insert a “; “ between “methods” and “however “ Section 3.2 

perhaps figure 5 could be inserted under this section, not above. But evaluate you how is the better way in your opinion

section 5 
 you cite two references regarding the morphological effect of essential oils on bacteria; could you cite also one or more works related to the effect of essential oils on morphology of fungi? line 171, please could you explain better this concept? “possibly attributed to less 
hyphase 
”

Author Response

Dear reviewers:

We greatly appreciate all the critiques and comments from you. Those comments are extremely helpful for us to improving our paper, and they provide valuable guidance for our future study. According to these comments, we have carefully improved our manuscript, and all the revisions are highlighted in red in the text. Please see below point-by-point responses to the reviewers’ comments:

Response to Reviewer 2’s comments:

Question 1:

Abstract

Please, check the term “comiditities” 

Introduction:

line 27 Prunus, instead of prunus

Line 52 and line 54: in vitro should be written in italics

Line 64: limits instead of iimits

Results and discussion

Line 91 antifungal instead of Antifungal

Line 94: delete “are”

Line 102: please, insert a “; “ between “methods” and “however “ Section 3.2 

perhaps figure 5 could be inserted under this section, not above. But evaluate you how is the better way in your opinion

R: Thanks for your valuable comment. As suggested, we have carefully corrected the errors you mentioned in the comments.

Abstract, line 23 “comiditities” is changed to "commodities".

Introduction, line 27 "prunus" is changed to "Prunus".

                   line 52 and line 54 “in vitro” is written italics.

                    line 64 "iimits" is changed to "limits.

Results and discussion, Line 91 "Antifungal" is changed to "antifungal".

                                  line 94 "are" is deleted.

                                line104 (revised version) ";" is inserted between                                              “methods” and  “however"  Section 3.2 
.

                                 figure 5 is inserted under this section.

Question 2: section 
 you cite two references regarding the morphological effect of essential oils on bacteria; could you cite also one or more works related to the effect of essential oils on morphology of fungi? line 171, please could you explain better this concept? “possibly attributed to less 
hyphase 
”

R: Thanks for your valuable suggestion. As suggested, we cite one work related to the effect of essential oils on morphology of fungi as reference 34 in line 166(revised version), and line165 (revised version) "bacteria" is changed to "microorganism"

 line173 (revised version) "less hyphase" is changed to "less yield of mycelia " which means growth quantity of mycelium.

Reviewer 3 Report

The study comes up with an interesting hypothesis. The authors showed efficacy of seven essential oils in suppressing Collelotrichum gloeosporioides identified from sweet cherry. The manuscript focuses an interesting topic that is worth to be published. The manuscript is well organized, but moderate english changes required.

Author Response

Dear reviewer:

We greatly appreciate all the critiques and comments from you. Those comments are extremely helpful for us to improving our paper, and they provide valuable guidance for our future study. According to these comments, moderate English have been improved, and all the revisions are highlighted in red in the text. Please see below some editings of English language and style.

Abstract, line 23 “comiditities” is changed to "commodities".

Introduction, line 27 "prunus" is changed to "Prunus".

                   line 52 and line 54 “in vitro” is written italics.

                    line 64 "iimits" is corrected to "limits.

Results and discussion, Line 91 "Antifungal" is changed to "antifungal".

                                  line 94 "are" is deleted.

                                 line104 (revised version) ";" is inserted between “methods”                                   and “however" Section 3.2 
.

Reviewer 4 Report

The paper by Wang et al. studies the effect of several commercial essential oils on the sweet cherry rot caused by fungi. 

This study may not be the most original or innovative research but the paper is very clearly written and this study is very well designed.

It is therefore very hard for the  reviewer to voice any concerns about this paper or  to add anything to improve the paper.

Therefore it is my opinion that this paper can and should be published as it is.

Author Response

Dear reviewer:

We greatly appreciate all the critiques and comments from you. Those comments are extremely helpful for us to improving our paper, and they provide valuable guidance for our future study.

Round 2

Reviewer 1 Report

Even if my first evaluation was "reject", the Authors respond to all my criticisms and i am satisfied.

Now the revised paper can be accepted